# Activity of Cerebellar Nuclei Neurons Correlates with ZebrinII Identity of Their Purkinje Cell Afferents

**DOI:** 10.3390/cells10102686

**Published:** 2021-10-07

**Authors:** Gerrit C. Beekhof, Simona V. Gornati, Cathrin B. Canto, Avraham M. Libster, Martijn Schonewille, Chris I. De Zeeuw, Freek E. Hoebeek

**Affiliations:** 1Department of Neuroscience, Erasmus Medical Center, 3015 AA Rotterdam, The Netherlands; g.beekhof@erasmusmc.nl (G.C.B.); simona.gornati@gmail.com (S.V.G.); 2Netherlands Institute for Neuroscience, Royal Academy of Arts and Sciences (KNAW), 1105 BA Amsterdam, The Netherlands; c.canto@nin.knaw.nl; 3Edmond & Lily Safra Center for Brain Sciences (ELSC), Department of Neurobiology, Institute of Life Sciences, The Hebrew University, Jerusalem 91904, Israel; avi.libster@gmail.com; 4Department for Developmental Origins of Disease, Wilhelmina Children’s Hospital, Brain Center, University Medical Center Utrecht, 3584 EA Utrecht, The Netherlands

**Keywords:** cerebellar nuclei, action potential firing, development, ZebrinII, morphology

## Abstract

Purkinje cells (PCs) in the cerebellar cortex can be divided into at least two main subpopulations: one subpopulation that prominently expresses ZebrinII (Z+), and shows a relatively low simple spike firing rate, and another that hardly expresses ZebrinII (Z–) and shows higher baseline firing rates. Likewise, the complex spike responses of PCs, which are evoked by climbing fiber inputs and thus reflect the activity of the inferior olive (IO), show the same dichotomy. However, it is not known whether the target neurons of PCs in the cerebellar nuclei (CN) maintain this bimodal distribution. Electrophysiological recordings in awake adult mice show that the rate of action potential firing of CN neurons that receive input from Z+ PCs was consistently lower than that of CN neurons innervated by Z– PCs. Similar in vivo recordings in juvenile and adolescent mice indicated that the firing frequency of CN neurons correlates to the ZebrinII identity of the PC afferents in adult, but not postnatal stages. Finally, the spontaneous action potential firing pattern of adult CN neurons recorded in vitro revealed no significant differences in intrinsic pacemaking activity between ZebrinII identities. Our findings indicate that all three main components of the olivocerebellar loop, i.e., PCs, IO neurons and CN neurons, operate at a higher rate in the Z– modules.

## 1. Introduction

The cerebellum integrates inputs from sensory, motor, cognitive and limbic systems in dedicated parts of its cortex and nuclei [1]. Cerebellar defects that span several lobules are, therefore, likely to result in complex disorders affecting, for instance, social behavior, sensorimotor integration and language [2]. Based on the connectivity of the pre-cerebellar afferent systems, i.e., mossy fibers and climbing fibers, as well as the cellular composition, the cerebellar cortex and cerebellar nuclei (CN) can be divided in modules. Although there are unifying theories on how the crystalline cerebellar architecture allows processing of diverse types of information, all of which enter as action potential (AP) firing patterns with specific spatiotemporal patterns [3], it remains to be elucidated how the modular organization of the olivo-cerebellar system correlates to the actual output of the cerebellum, i.e., the CN firing pattern.

Various factors are known to influence the CN firing patterns. CN neurons are characterized by pacemaking activity, i.e., fire APs even in absence of any synaptic inputs [4] and receive excitatory mossy fiber and climbing fiber input via axon collaterals. Additionally, CN neurons receive inhibitory input from Purkinje cells (PCs), the axons of which synapse on the perisomatic membrane [5]. Despite the dense PC convergence onto CN neurons (40:1 in adult murine brain; [6]), CN neurons are known for their extensive dynamic range in firing frequency, which is particularly evident from recordings in awake preparations [7,8,9,10,11,12,13]. The precise impact of the excitatory and inhibitory afferents has been studied in detail in silico, in vitro and in vivo [6,14,15,16,17,18], but without attention for the impact of the modular organization of the cerebellum.

Immunohistochemical stainings for several markers, of which ZebrinII (aldolase-C) is the most frequently used, clearly reveal that the CN can be divided into two domains, a ZebrinII-positive (Z+) and a ZebrinII-negative (Z–) domain, based upon the density of Z+ and Z– PC axons [19,20]. The functional relevance of the ZebrinII identity of individual CN has recently been identified. The lateral, posterior interposed and caudal portion of the medial CN are mostly innervated by Z+ PCs, whereas the anterior interposed and rostral portion of the medial CN are mostly innervated by Z– PCs [21]. Recent studies in adult mice have shown that the Z+ PCs fire APs on average at lower frequencies than Z– PCs [22,23], which has been linked to various behaviors. For instance, eye blink conditioning is encoded in Z– anterior interposed nucleus, which receives innervation from the portion of the cerebellar cortex in which Purkinje cells on average fire at a higher frequency [24]. Also in the medial CN the ZebrinII identity was recently shown to differ for various functional domains, in that motor, positional, autonomous and vigilance information is encoded by various regions in the medial CN that receive input from Z+ or Z– Purkinje cells [25]. To investigate whether the input from Z+ or Z– Purkinje cells has an effect on the firing frequency of CN neurons, we established the firing patterns of CN neurons in Z+ and Z– nuclei in various developmental stages and assessed the impact of synaptic inputs. 

## 2. Materials and Methods

### 2.1. Subjects

For in vivo extracellular recordings in the CN, we used a total of 62 mice, aged between postnatal day (P)12 to P133. We used both males and females and selected the pups randomly from a litter, and used either *Slc1a6-EGFP* [26] or C57BL/6J mice to record CN neurons. We recorded 111 CN neurons in 18 *Slc1a6-EGFP* and 293 CN neurons in 44 C57BL/6J mice. The *Slc1a6-EGFP* mice were used to shorten the histological procedure, since the ZebrinII pattern is already fluorescent. All recording locations where stained with biocytin or Evans Blue, therefore we did not use any mice for both in vivo and in vitro recordings. For in vitro whole-cell recordings in CN we used a total of 22 *Slc1a6-EGFP* mice. The *Slc1a6-EGFP* mice were used to visualize the ZebrinII identity of the CN at the patch setup. We calculated the difference between *Slc1a6-EGFP* and C57BL/6 recordings, but found no significant difference between any parameter (data not shown). 

The *Slc1a6-EGFP* mice were bred in-house by crossbreeding with C57BL/6 mice. The C57BL/6J mice used for experiments were obtained from a time-pregnant female imported from the vendor (Charles River, Wilmington, MA, USA, or Janvier Labs, Le Genest-Saint-Isle, France). All experiments were performed in accordance with the European Communities Council Directive. Protocols were reviewed and approved by the Dutch national experimental animal committees (DEC) and every precaution was taken to minimize stress, discomfort and the number of animals used. 

### 2.2. Surgery for In Vivo Awake Recordings

Mice were subcutaneously injected with buprenorphine (0.015 mg/kg) (RB Pharmaceuticals Ltd., Slough, UK) and Rimadyl cattle (5 mg/kg) (Zoetis, Parsippany, NJ, USA) 60 min before surgery. Mice were anesthetized with isoflurane (3% in 0.4 L/min O_2_ for induction and for maintenance 0.5–1.3% in 0.2–0.4 L/min O_2_) (TEVA Pharmachemie, Haarlem, The Netherlands). During the surgery the temperature was maintained at 37 °C using a heating pad and anal probe in an automated feedback system. Before and after shaving of the skin 2% lidocaine (AstraZeneca, Cambridge, UK) was applied. To expose the skull the skin was opened over the rostro-caudal midline. The skull was covered with a layer of Optibond (Kerr, Salerno, Italy) for stability and 5 holes were drilled using a high speed, diamond-tipped drill (Foredome, Bethel, CT, USA). To obtain ECoG signals, five pure silver ball-tipped electrodes (custom-made from 0.125 mm diameter silver wire; Advanced research materials LTD, Eynsham, Oxford, UK) were placed on the meningeal layer of the dura mater. Two silver electrodes were positioned bilateral above the primary cortex (M1: 1 mm rostral; 1 mm lateral; relative to Bregma), two were placed above the primary sensory cortex (S1: 1 mm caudal; 3.5 mm lateral relative to Bregma), and one in the interparietal bone (1 mm caudal; 1 mm lateral relative to Lambda). UV-sensitive composites, a layer of Optibond (Kerr, Salerno, Italy) and Charisma or Charisma Flow (Heraeas Kulzer, Hesse, Germany), were used to fixate the silver electrodes and the head-fixing pedestal, consisting of two M1.4 nuts. To obtain extracellular recordings a craniotomy was made in the occipital bone, which was temporarily closed with Kwik-Cast sealant (World Precision Instruments Inc., Sarasota, FL, USA) to prevent cooling of the brain. At the end of the surgery the mice received 0.1–0.2 mL saline intraperitoneal injection for hydration.

### 2.3. In Vivo Awake Extracellular Recordings

Within 2 h of the start of the surgical procedure the mice were relocated to the setup where the body temperature was supported via a feedback-controlled heating pad. We evaluated the state of the mice using the behavior (spontaneous whisking, response to auditory stimuli) and ECoG (presence of slow-wave, high-amplitude activity indicating drowsiness, or fast, low-amplitude activity indicating alert state). ECoG and extracellular recordings were sampled at 20 kHz (setup 1: Digidata 1322A, Molecular Devices LLC., Axon instruments, Sunnyvale, CA, USA), amplified, and stored for offline analysis (CyberAmp 380 and Multiclamp 700A, Molecular Devices) or at 50 kHz (setup 2: ECoG: adapted MEA60, Multichannel system, Reutlingen, Germany; extracellular: Multiclamp 700B amplifier with a DigiData 1440; Molecular Devices). Single-unit recordings started 2 h or more after the termination of isoflurane application and only when the ECoG appeared normal for an awake mouse (evaluated by the experimenter from raw traces), free from NREM and REM sleep waves in the ECoG [27] and display alert whisking behavior. Next, cells were recorded using borosilicate glass capillaries (Harvard apparatus, Holliston, MA, USA) with 0.5–1.0 µm tips and a resistance of 6–12 MΩ. Glass pipettes were filled with internal solution containing (in mM): 9 KCl, 3.48 MgCl_2_, 4 NaCl, 120 K^+^-Gluconate, 10 HEPES, 28.5 Sucrose, 4 Na_2_ATP, 0.4 Na_3_GTP (Sigma-Aldrich, Merck KGaA, Darmstadt, Germany) in total pH 7.25–7.35, osmolarity 290–300 mOsmol/Kg; and 1% biocytin or 0.5% Evans Blue (Sigma-Aldrich, Merck KGaA, Darmstadt, Germany). We used an internal recording solution to easily dissolve biocytin or EvansBlue. At the recording location biocytin was released with iontophoresis with 1 s pulses of 4 µA for 3 min (custom-built device, Erasmus MC, Rotterdam, The Netherlands), or Evans blue was injected with pressure. In our analysis, we included only cells that we could identify the injection spot by the recording location, see Figure 1A (right panel). We determined the specific subnuclei using confocal images (see Appendix A), the confocal images were compared to both the Paxinos and Franklin atlas [28], and the previously published ZebrinII staining in the CN [20] (Figure 1, Table 1). This process was at least performed by two independent observers; in case the observers’ classification of the ZebrinII identity did not match, the neuron was discarded from further analysis.

### 2.4. Slice Preparation for In Vitro Whole-Cell Recordings

Adult mice (>P90) were isoflurane-anesthetized before decapitation, their brain quickly removed and placed in warm (~34 °C) artificial cerebrospinal fluid (aCSF) containing the following (in mM): 123 NaCl, 2.5 KCl, 1 MgCl_2_, 1.3 NaH_2_PO_4_, 26 NaHCO_3_, 10 glucose, 2 CaCl_2_, bubbled with 95%O_2_/5%CO_2_, pH 7.4 [29]. Coronal slices (250 µm) of cerebellar tissue including CN were cut using a vibratome (VT1200S, Leica Biosystems, Wetzlar, Germany) with a ceramic blade (Campden Instruments Ltd., Manchester, UK). Directly after slicing the cerebellar slices were transferred to a recovery bath, and were incubated in oxygenated (bubbled with 95% O_2_ and 5% CO_2_) aCSF and maintained at 34 ± 1 °C. After 30 min the slices were transferred to a recording chamber and maintained at 34 ± 1 °C under continuous perfusion with the oxygenated physiological solution. 

### 2.5. In Vitro Whole-Cell Recordings

For all recordings, slices were bathed in 34 ± 1 °C aCSF (bubbled with 95% O_2_ and 5% CO_2_). Whole-cell patch-clamp recordings were performed using an EPC-10 amplifier (HEKA Electronics, Lambrecht, Germany) for 20–60 min and digitized at 20 kHz. Whole-cell recordings were obtained using borosilicate pipettes (4–6 MΩ) filled with internal solution containing (in mM): 120 K-gluconate, 6 NaCl, 10 HEPES, 1 EGTA-KOH, 0.1 CaCl_2_, 4 Mg-ATP, 0.4 Na-GTP, 2 KCL, 14 Creatine phosphate TRIS, 2 MgCl_2_ (pH 7.36, osmolarity 290 mOsmol/Kg). After breaking the gigaseal, the spontaneous activity was first recorded for at least 2 min. Recording pipettes were supplemented with 1 mg/mL biocytin to allow histological staining (see below). All recordings were performed in the presence of picrotoxin (100 µM, Sigma-Aldrich, Merck KGaA, Darmstadt, Germany) to block GABA_A_-receptor-mediated inhibitory postsynaptic currents (IPSCs) and glycine receptors, NBQX (10 µm, Tocris, Bristol, UK) and 10 μM APV (10 µm, Tocris) to block respectively AMPA and NMDA receptors.

### 2.6. Measurement of In Vitro Electrophysiological Parameters

Current-clamp traces were acquired using Patchmaster software (HEKA Elektronik Dr. Schulze GmbH, Lambrecht, Germany) and stored for offline analysis. Immediately after breaking the gigaseal, the resting membrane potential (Vrest) was recorded as well as the spontaneous AP firing. Vrest was calculated as the mode of the membrane potential during the first 1000 ms of a I = 0 pA current-clamp recording. These recordings lasted 120 s each and were used to calculate the spontaneous AP firing pattern. Input resistance (RI) and series resistance (Rs) were recorded in voltage-clamp mode by −5 or −10 mV voltage steps. Recordings were discarded from further analysis if RI or RS varied by >25% over the course of the experiment. For analysis of evoked APs, the first APs fired by each cell in response to a series of increasing depolarizing current steps was isolated and analyzed using a custom build Matlab code (Maltab 2016, Mathworks, Natick, MA, USA). The AP threshold was calculated as the potential at which the second derivative peaked (d^2^V/dt^2^) [30]. The baseline for analyzing the amplitudes of the AP and the fast after-hyperpolarization (fAHP) were calculated relative to the mode of 10 ms of the trace (from −15 to −5 ms before the AP peak). The rise and decay time were calculated using the AP threshold as start point using 10–90% of the total rise and decay period, the half-width was also calculated taking the AP threshold as the start of the AP. For current-frequency (I-F) plots we calculated the average firing frequency for each level of injected current. We calculated the rheobase as the first level of current injection (50 pA steps) that evoked an AP. The frequency adaptation was calculated from selected traces with an average firing rate of ~40 Hz for both Z+ and Z– cells and normalized all interspike intervals to the initial interval. The recording traces with current injections were repeated three times and values were averaged per cell.

### 2.7. Immunofluorescence for In Vitro Recordings

Slices were placed in 4% PFA (in 0.12 M phosphate buffer (PB)) for at least 24 h. Subsequently, slices were transferred into 0.1 M phosphate buffered saline (PBS), rinsed with PBS three times for 10 min and incubated for 1 h at room temperature (RT) in blocking solution. Thereafter, the slices were rinsed three times for 10 min and incubated for 2 h with Streptavidin-Cy3 (1:200, Jackson Immuno Research Inc., West Grove, PA, USA) diluted in PBS containing 2% normal horse serum and 0.4% triton. Finally, slices were rinsed in PBS, mounted with Vectashield (Vector laboratories, Burlingame, CA, USA) and imaged with a LSM 700 confocal microscope (Carl Zeiss Microscopy, LLC., Thornwood, NY, USA). 

### 2.8. Fluorescence Microscopy for 2D Sholl Analysis 

Recorded neurons were labeled with biocytin. Epifluorescent tile images were obtained using a 20×/0.30 NA (air) objective and a LSM 700 microscope (Carl Zeiss). The position of labeled neurons was confirmed using a stereotactic atlas. To determine the dendritic arborization of biocytin filled cells, we imaged using a 40×/1.3 numerical aperture (NA) oil-immersed objective to acquire a stack of images with 0.5× digital zoom and a voxel size of 313 nm width × 313 nm length × 300 nm depth for a 2D Sholl analysis by a Sholl analysis macro implemented in FIJI [31] software. For preprocessing, stacks with excessive background signal were excluded from further analysis. Subsequently the maximum projection of each image was thresholded in FIJI, and the dendritic arborization was measured in concentric shells of 10 µm distance starting with 15 µm distance from the center of the soma.

### 2.9. Immunofluorescence for In Vivo Recordings

At the end of the in vivo recordings mice were sacrificed with an overdose of pentobarbital and transcardially pre-rinsed with PBS followed with a solution of 4% paraformaldehyde in 0.1 M PB. The brain was removed and post-fixed for two hours at room temperature (RT) in this paraformaldehyde solution, and then placed overnight at 4 °C in a solution of 10% sucrose in 0.1 M phosphate buffer. Brains were embedded in a 0.1 M PB with 12% gelatin and 10% sucrose, the embedded brain was fixed for two hours in 10% formaldehyde and 30% sucrose solution of 0.1 M PB. Then it was placed in a 0.1 M PB with 30% sucrose overnight at 4 °C. Brains were sliced in 40 or 100 µm thick slices on a freezing microtome. A standardized immunochemistry protocol was used and all slices from C57BL/6 mice were stained with the primary aldolase-C (goat, 1:1000, Santa Cruz Biotechnology Inc., Dallas, TX, USA) for four days at 4 °C, and for two hours at RT for the secondary antibody anti-goat (1:200, Jackson Immuno Research). Note that the aldolase-C staining is complementary with the *Slc1a6-EGFP* expression pattern [26] and that ZebrinII is the same as aldolase-C [32]. In case biocytin was used, slices were also stained with streptavidin (1:200, Jackson Immuno Research). All slices received a nuclear DAPI staining for 10 min, and were mounted on cover slips from a chromium(III) potassium sulfate solution and then covered with a microscope slide with mowiol.

### 2.10. Acquisition of Confocal Images

Wide-field fluorescent tile scan images were acquired with a LSM 700 (Carl Zeiss) or a SP5/SP8 (Leica Microsystems) confocal laser scanning microscope. The tile scans of the injection spots were made with a 10× objective, 20% overlap and online-stitched. Only recorded neurons that were located within 300 µm from the center of the injection area (identified by the Evans blue or biocytin staining) were analyzed. If the injection center was near the border between Z+ and Z– domains and at least one of the experimenters G.B.C, S.V.G. and/or F.E.H. were uncertain about the definitive location, the recording was excluded from analysis. 

### 2.11. Spike Analysis

For spike analysis of the CN neurons (n = 404 cells) recorded in vivo we included only cells with a recording length of at least 90 s (duration: 214 ± 160 s). In our analysis we included only the cells of which we could trace the recording location by the use of the injection spot. We used a Matlab (Mathworks) code to detect spikes using threshold and principal component analysis [33]. We analyzed the regularity of AP firing patterns using the coefficient of variance (CV) and CV2. CV is the variation in interspike intervals (ISI): CV = standard deviation ISI/mean ISI(1)

CV2 represents the variance on a spike-to-spike base and is less sensitive for a single outlier compared to CV [34]:CV2 = (2 × |ISI_n+1_ − ISI_n_|)/(ISI_n+2_/ISI_n_)(2)

The mean instantaneous firing frequency (miFF) is the average firing frequency (FF) on a spike-to-spike basis,
miFF = mean ((1/ISI_n_) + (1/ISI_n+1_))(3)

The burst index reflects the fraction of all spikes that were part of a ‘burst’, which we defined by at least three spikes at ≥100 Hz followed by a pause of 50 ms (‘burst index 50 ms’) or 100 ms (‘burst index 100 ms’).

### 2.12. Statistical Analysis

Using GraphPad Prism8.3.1 (GraphPad Software, San Diego, CA, USA) we ran the appropriate statistical comparisons between groups and subgroups, the outcome of which is represented in designated Appendix A. We defined *p* < 0.05 as a significant difference. Summarized data are represented as mean with ± standard error of the mean (SEM) unless stated otherwise.

## 3. Results

### 3.1. Firing Frequency Differs between ZebrinII Domains in Cerebellar Nuclei during Adulthood

To confirm the dichotomous labelling of the CN that was shown in the rat [21], we evaluated the presence of a clear separation between CN innervated densely by Z+ PC axons and by Z– PC axons assessing the eGFP labelling of ZebrinII (aldolase-C) in the CN in *Slc1a6-EGFP* mice [22,35,36]. Indeed, we found that the Z+ domains of the CN encompass the posterior interposed, lateral and the medio-ventral portion of the medial nuclei, whereas the anterior interposed and the latero-dorsal portion of the medial nuclei are innervated predominantly by Z– fibers (Figure 1A; Appendix A).

To evaluate the average firing frequency of cerebellar nuclei neurons located in the Z+ and Z– domains (hereafter referred to as Z+ and Z– CN neurons, respectively), we performed extracellular in vivo recordings throughout the CN in awake, adult mice (>P90). A total of 97 of the recorded neurons was confirmed to be located in the CN by histological verification (see Section 2) (Figure 1A). The CN cells showed disparity in firing patterns (Figure 1B; Appendix A) as a higher firing frequency was observed in Z– CN neurons compared to Z+ CN neurons (Z+: 53.9 ± 3.5 Hz; Z–: 68.7 ± 4.7 Hz; *p* = 0.0107, unpaired *t*-test; Figure 1C; Appendix A). The regularity of AP firing was quantified by calculating the CV and the CV2; we found that the regularity was not significantly different between neurons recorded in Z+ or Z– CN (CV: Z+: 1.25 ± 0.27; Z–: 1.02 ± 0.29; *p* = 0.6211, Mann–Whitney U test; CV2: Z+: 0.48 ± 0.02; Z–: 0.48 ± 0.03; *p* = 0.9957, Mann-Whitney U test; Figure 1D_1_,D_2_ respectively; Appendix A). Because of the higher firing frequency of Z– cells, we investigated whether the Z– cells were more prone to burst firing. We found that the fraction of burst firing in Z– CN neurons is not significantly different from Z+ CN neurons (Z+: 0.0027 ± 0.0018; Z–: 0.0008 ± 0.0003; *p* = 0.7041 Mann–Whitney U test; Figure 1E; Appendix A). Hence, we investigated the interspike interval distributions, which can be described by the skewness, a parameter for the asymmetry of the distribution, and the kurtosis, for the ‘tailedness’ of the distribution. We found no significant difference in the skewness (Z+: 11.3 ± 3.7; Z–: 7.5 ± 2.7; *p* = 0.92 Mann–Whitney U test; Figure 1F; Appendix A) and kurtosis (Z+: 984 ± 629; Z–: 380 ± 289; *p* = 0.86 Mann–Whitney U test; Figure 1G; Appendix A) between Z+ and Z– CN neurons. 

### 3.2. No Differences in Firing Frequency between ZebrinII Domains in Cerebellar Nuclei (CN) during Development

The firing frequency of Z+ and Z− PCs starts to differentiate from P12 [23], and the ZebrinII pattering in PCs is complete around P12-15 [37,38]. To determine if a similar timeline is present for CN neurons, we recorded the activity of Z+ and Z− CN neurons at different developmental stages. Starting from P12, we found a gradual increase in the firing frequency for both Z+ and Z− CN neurons (Figure 2A,B; Appendix A). However, when comparing the firing patterns of Z+ and Z− CN neurons during the different developmental stages, we found that at none of the studied ages, i.e., P12-14, P15-17 and P18-P20, a significant difference. Also when we evaluated the firing patterns we found no significant differences, except for the a more irregular firing pattern in Z+ domains compared to Z− domains at P15-17 (CV: Z+: 1.77 ± 0.21; Z−: 1.00 ± 0.13; *p* = 0.0082, Kruskal–Wallis test; CV2: Z+: 0.69 ± 0.03; Z−: 0.55 ± 0.05; *p* = 0.0267; Brown-Forsythe and Welch ANOVA; Figure 2C_1_,C_2_,Appendix A). The burst index of spikes fired within a burst that is followed by a pause of ≥50 ms is significantly different for P21-24 (Z+: 0.0026 ± 0.0006; Z–: 0.0003 ± 0.0001; *p* = 0.0052 Kruskal–Wallis test; Figure 2D; see also Appendix AB; Appendix A) and P30-P40 (Z+: 0.0032 ± 0.0013; Z–: 0.0001 ± 0.0001; *p* = 0.0054 Kruskal-Wallis test; Figure 2D; see also Appendix A). In all other ages there were no significant differences between the firing pattern parameters (all *p*-values > 0.05; Figure 2, see also Appendix A). These findings indicate that the difference between the firing frequency of Z+ and Z− CN neurons that we recorded in vivo in adult mice (>P60) appears only after P40.

### 3.3. Similar Excitability and Pacemaker Activity of Cerebellar Nuclei Neurons in ZebrinII Domains

As CN cells are known to fire APs in the absence of synaptic input [4], we investigated whether the lower firing frequency of Z+ CN neurons compared to the Z– CN neurons recorded in vivo is due to a difference in the spontaneous activity of CN neurons. To do so, we performed whole-cell patch-clamp recordings of CN neurons from coronal cerebellar slices of adult (>P90) mice. We identified CN neurons with a relatively large soma as the big glutamatergic cells [39] and recorded the resting membrane potential and AP firing patterns in the presence of AMPA-, NMDA-, GABA_a_-. glycine-receptor blockers (NBQX, APV and PTX, respectively; Figure 3A). We did not observe a difference in the resting membrane potential (V_rest_: Z+: −48.2 ± 1.0 mV; Z–: −47.2 ± 0.7 mV, *p* = 0.4633: unpaired *t*-test; Figure 3B; Appendix A). The firing frequency was also not significantly different between the two groups (Z+: 70.8 ± 9.1 Hz; Z–: 83.9 ± 10.0 Hz; *p* = 0.4363, Mann–Whitney U; Figure 3C; Appendix A). Moreover, the regularity of AP firing did not show any trend (CV: Z+: 0.17 ± 0.04; Z–: 0.13 ± 0.02; *p* = 0.7692, Mann–Whitney U test; CV2: Z+: 0.05 ± 0.01; Z–: 0.05 ± 0.01; *p* = 0.8698; Mann–Whitney U test; Figure 3D_1_,D_2_; Appendix A). 

Next, we studied AP firing dynamics and focused on the firing frequency versus injected current (F-I) relationship, which provides information related to the excitability of a neuron and how stable the resulting AP firing pattern is. We found that the rheobase was not different between the two groups; CN neurons from both Z+ and Z– domains showed AP firing in response to the smallest current step used (50 pA of depolarizing current; Figure 3E,F; Appendix A). Also the frequency of the evoked APs did not show a significant effect of the ZebrinII identity (*p* = 0.2687, 2-way ANOVA, Appendix A). When testing the AP adaptation, we found that there was a significant effect on the number of interspike intervals (*p* < 0.0001), but no effect of ZebrinII-identity (*p* = 0.7614, mixed-effect analysis; Figure 3G; Appendix A). In addition, we also analyzed the shape of the first evoked AP fired in response to series of depolarizing current steps of increasing amplitudes (from −100 pA to 800 pA). We found no significant differences in the AP threshold, half-width, rise time, decay time, peak amplitude and after hyperpolarization (all *p*-values > 0.05; Appendix A). 

### 3.4. Morphology of Cerebellar Nuclei Neurons in ZebrinII Domains

Following immunohistochemical staining, we reconstructed the morphology and location of the patched neurons (Figure 4A; Appendix A). We found that the average surface of the somata was not different between Z+ (306.4 ± 16.7 µm^2^) and Z– (342.1 ± 30.2 µm^2^; *p* = 0.3121, Welch *t*-test; Figure 4B_1_; Appendix A). The dendritic branching was investigated using 2D Sholl analysis and revealed that although there was a significant effect of distance to soma (*p* < 0.0001), the dendritic arbor complexity decreased with increasing distance from the soma. There was no effect of ZebrinII identity (*p* = 0.3675) nor a consistent difference at the intersections between Z+ and Z– (mixed-effects model; Figure 4B_2_; Appendix A).

## 4. Discussion

Here we investigated the differences in spiking activity of CN receiving inputs from Z+ and Z– PCs in mice. Our in vivo recordings of AP firing showed that the average firing frequency of CN neurons was higher in the Z– than in Z+ domain, but that this difference could only be confirmed for the adult stage, not during the postnatal stages between P12-P40. To investigate whether the intrinsic pacemaking activity of CN neurons contributes to this difference, we performed in vitro recordings of AP firing in the presence of neurotransmitter blockers. In the absence of input we found no significant difference between the spontaneous and evoked spiking patterns between the CN neurons in Z+ and Z– domains. These findings suggest that the difference in firing pattern between Z+ and Z– domains in CN are absent during development and, at least partially, caused by synaptic afferents. 

Previous studies showed that PCs with a Z– identity on average have a higher firing frequency than Z+ PCs [22,23,40,41]. Given the inhibitory effect of PC afferents on CN neurons, we expected to observe a lower firing frequency in the Z– domain of the CN than in the Z+ domain. However, conversely we found that the firing frequency of Z– CN neurons is higher than that of Z+ CN neurons. These counterintuitive findings might be, at least partially, due to four reasons: firstly, the heterogeneous population of neurons of the cerebellar nuclei [42]. Although we used extracellular recording electrodes and thus presumably recorded the cells with a relatively large soma diameter and continuous AP firing (see also [7]), it may still be that we recorded from various neurochemical subzones [19]. Secondly, a potential difference in the number of synaptic inputs between CN neurons in the different ZebrinII domains, which extends beyond the PC input and that of local inhibitory interneurons to extracerebellar sources of excitatory input from mossy fiber and climbing fiber collaterals [17,18,43]. Thirdly, a potential difference in intrinsic firing rate between Z– and Z+ CN neurons. The higher firing rate in Z– PCs in vivo was reproduced by recordings in vitro, even in small sample sizes (see Figure 5C in [22]). In our in vitro CN neuron recordings we did not observe a robust difference, but that potential difference might have been obscured by the large variability in firing rates in vitro. Fourth, and finally, the potentially different levels of synchrony in PC AP firing between the microzones that receive input from electrotonically- and functionally-coupled IO neurons [44] and possibly innervate different ZebrinII domains of the CN. Person and Raman (2011) demonstrated that the synchrony of the inhibitory impact of PCs determines the generation of CN APs [6]. Which of these four factors affects the difference in Z+ and Z– firing rates in adult stages remains unknown.

Apart from PC inputs, also the excitatory afferents from mossy and climbing fiber collaterals control CN spiking patterns. For mossy fiber inputs it was shown that selective stimulation can evoke CN AP firing [45], a feature that can be modulated by the level of PC synchrony [46]. Also climbing fiber collaterals have been identified as drivers of CN AP firing [17]. Due to the electrotonic coupling of IO neurons, the synchronicity of climbing fiber activity can be effectively coupled, substantiating the potential role of climbing fiber-mediated CN spiking. A previous in vivo study indicates that in anesthetized mice ~15% of CN APs is indeed preceded by putative climbing fiber inputs [47]. Whether a similar percentage applies to our recording conditions is unclear, but we assume that also in our in vivo dataset part of CN spiking is driven by excitatory inputs. 

Although the firing frequencies from our in vivo awake CN recordings is in the range of previous findings (e.g., [7]) we cannot exclude that the use of the anesthetics isoflurane and buprenorphine prior to the surgery has had an effect on AP firing. The continued effect of isoflurane on PC firing patterns after the administration had been ceased was experimentally determined to have disappeared after 8 minutes [48]. For CN recordings we found the impact of isoflurane to be even shorter [7], indicating that for the current dataset the use of isoflurane will have had a very limited influence, since it was terminated at least two hours prior to the actual recording of neural activity. In contrast, buprenorphine has a half-life time in adult mice of ~3 h and could potentially have influenced our recordings [49]. One potential effect of buprenorphine administration can be a decrease of AP firing, as has been described for in vivo olfactory bulb recordings [50]. Although the exact effect of buprenorphine on cerebellar spiking patterns has yet to be determined, opiate receptors are present in cerebellar tissue [51]. What already is known is that buprenorphine has an impact on the firing frequency in regions that project to cerebellar neurons, such as the dopaminergic axons from the ventral tegmental area [52,53,54]. However, to the best of our knowledge these projections are not different between ZebrinII domains and thus may have a limited effect on the regional difference in cerebellar spiking patterns that we recorded in vivo. 

Our in vitro experimental design was initiated by acute slicing without cooling the tissue. This approach allowed us to record continuous spiking [29], which in the presence of AMPA, NMDA and GABA blockers, represented the intrinsic pacemaking activity of the CN neurons in ZebrinII domains. All the cells recorded showed high frequency firing rates as previously shown in large-soma glutamatergic neurons [4,17,39,55]. Moreover, we found by histological reconstruction that the recorded neurons in both Z+ and Z– domains had a relatively large soma and dendritic tree, comparable with previous data [17,39,56]. Our data, therefore, indicate that if we recorded from various cell types, this may have occurred in both ZebrinII domains. Another factor that may have influenced our recordings is the neurochemical heterogeneity of CN neurons [19]. Other than the ZebrinII domains, which in our experimental setup were identifiable by the high versus low PC expression level of EAAT4, the CN neurons can also be divided in several subclasses, some of which are prominent in individual CN. Future studies should identify the electrophysiological signatures of these neurochemically distinguishable CN populations and investigate the correlation with the ZebrinII identity of PC afferents. 

Our current experiments were performed from the second postnatal week, a time span for which it has been shown that GABA-mediated transmission in CN elicits inhibitory responses [57,58]. We and others found that during the first postnatal months the firing frequency of CN neurons increases over time [59,60,61], much alike the increase in PC firing [62]. Recently, our team revealed that Z+ and Z– PC firing already differs in postnatal mice, from P12 onwards to adulthood in both in vivo as in vitro recordings [23]. Why these differences are not matched in CN firing frequencies remains unknown. Potentially, a difference in the regularity or synchrony of postnatal Z+ and Z– PCs does not develop before adult stages. In adult mice the role of extrinsic input in PC firing rates is higher in Z– domains when compared with Z+ domains (see Figure 2-figure supplement 2A_2_ in [23]), contributing to the regional differences between the two subtypes of PCs. Since we only observed a higher firing rate in Z– CN neurons compared to Z+ CN neurons in vivo and not in vitro, the role of extrinsic input is leading in CN neuron ZebrinII-related firing.

Finally, it would be prudent to determine if there are differences between CN firing pattern of specific nuclei. For example, although the afferents to the anterior interposed have been studied in several species (as reviewed by [63]), we are not aware of publications that report differences in the firing patterns of this (or any other) CN. However, with the use of optogenetics and viral tracing techniques several recent studies have been able to specify regions within the interposed nuclei as functionally distinct [13,45,64,65]. We propose that the firing pattern of the ZebrinII domains are tuned to the specific functions, much like the diversification of the PC spiking patterns between ZebrinII bands [3,22,23,24,41,66,67]. Whether the downstream targets of CN neurons adhere to the distinct output patterns of ZebrinII domains remains a topic of further investigation. 

## 5. Conclusions

Baseline firing frequencies of Purkinje cells in the cerebellar cortex and neurons in the inferior olive show a dichotomous distribution in line with the ZebrinII identity of Purkinje cells. Here we show that the firing patterns of cerebellar nuclei neurons of awake alert adult mice adhere to the same distribution. 

## Figures and Tables

**Figure 1 cells-10-02686-f001:**
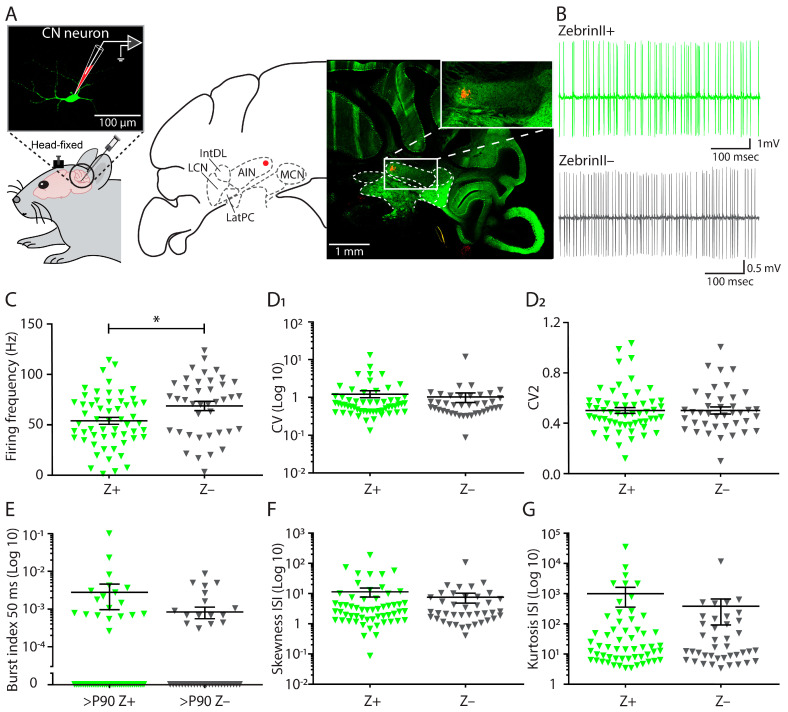
Cerebellar nuclei (CN) firing frequency differs between ZebrinII domains in vivo in the adult mouse. (**A**) (Left) An example picture of a fluorescently-labelled CN neuron, which was stained during whole-cell recording. (Right) Representative confocal image (and corresponding stereotactic atlas extracted from (Paxinos and Franklin, 2001) [28]) of a *Slc1a6-EGFP* cerebellar slice with a biocytin-labelled injection spot (red) at the recording location of an extracellularly recorded CN neuron in the anterior interposed nucleus. (**B**) Two representative example traces of CN neurons recorded in areas innervated by Z+ and Z– PC axons in adult mice (>P90). (**C**) Quantification of firing frequency (**C**), (**D_1_**) coefficient of variance (CV), and (**D_2_**) CV2 for Z– (gray, n = 40) and Z+ (green, n = 57) CN neurons recorded from 11 *Slc1a6-EGFP* mice. (**E**) The fraction of the total population of bursting spikes followed by ≥50 ms pause is displayed as the burst index. (**F**) The asymmetry of the distribution of interspike intervals (ISI) is represented by the skewness. (**G**) The ‘tailedness’ of the interspike interval (ISI) distribution is represented by the kurtosis. AIN = Anterior interposed nucleus; IntDL = dorsolateral hump of the interposed nucleus; LCN = lateral cerebellar nucleus; LatPC = lateral nucleus parvicellular part; MCN = medial cerebellar nucleus; PIN = posterior interposed nucleus. * denotes *p* < 0.05, each triangle represents a data point of single neurons. See Appendix A for all statistical data. Data are represented as mean ± SEM.

**Figure 2 cells-10-02686-f002:**
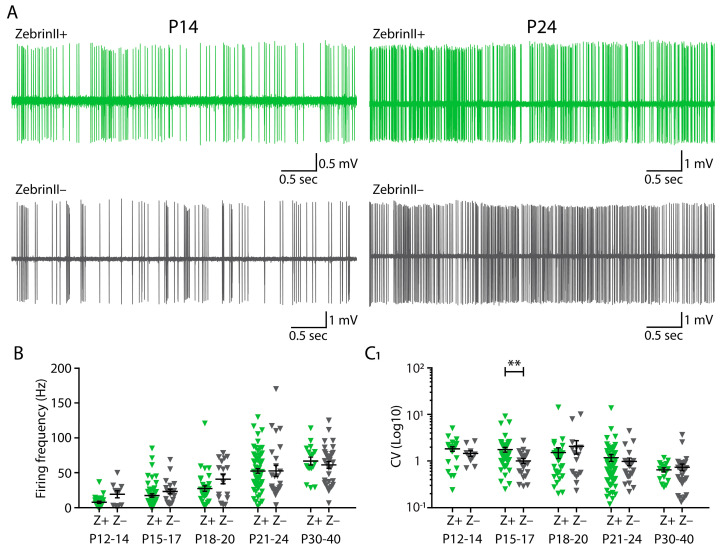
Firing frequency increases with age in both Z+ and Z– CN. (**A**) Two representative example traces for recordings from CN neurons from Z+ (green) and Z– (black) domains at P14 and P24. (**B**) Quantification of the mean firing frequency, (**C_1_**) CV, and (**C_2_**) CV2 for each age group (total Z+: n = 201 neurons from 52 mice; total Z–: n = 106 neurons from 52 mice). (**D**) The fraction of the total population of bursting spikes followed by a pause of ≥50 ms is displayed as the burst index for Z– and Z+ CN neurons in each age group. P = postnatal. * denotes *p* < 0.05, ** denotes *p* < 0.001, each triangle represents a data point of a single neuron. See Appendix A for all statistical data. Data are represented as mean ± SEM.

**Figure 3 cells-10-02686-f003:**
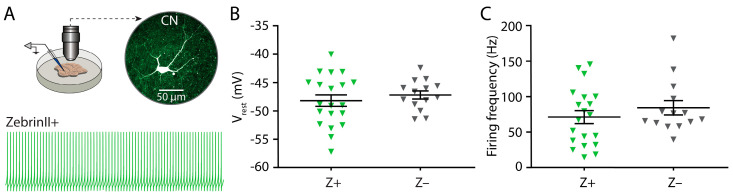
Firing frequency is similar in spontaneous spiking activity of Z+ and Z– CN neurons. (**A**) (Top) Immunofluorescence images of labelled CN neuron (white) following whole-cell recording with a biocytin-filled patch-electrode (green indicates *Slc1a6-EGFP*). (Bottom) Example traces of neurons recorded in vitro by whole-cell patch-clamp without injected holding current in Z+ and Z– nuclei. (**B**) Quantification of resting membrane potential (V_rest_), (**C**) firing frequency, (**D_1_**) coefficient of variance (CV) and (**D_2_**) CV2 for 20 Z+ and 14 Z– neurons recorded from 22 *Slc1a6-EGFP* mice. (**E**) Example traces of AP firing evoked by depolarizing current injection. (**F**) Average firing frequency evoked by steps of various current amplitudes (F-I curve). (**G**) Frequency adaptation depicted by interspike interval (ISI) length normalized to the first ISI. Each triangle represents a data point of a single neuron. See Appendix A for all statistical data. Data are represented as mean ± SEM.

**Figure 4 cells-10-02686-f004:**
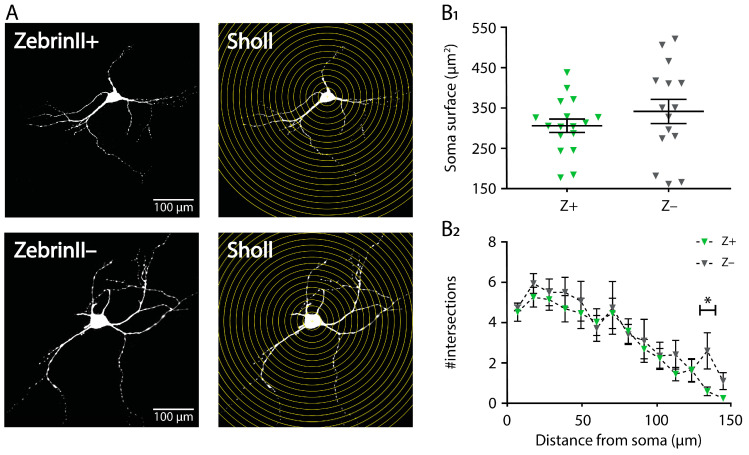
Z+ and Z– CN neurons do not differ in somatic and dendritic morphology. (**A**) Maximum projection of a confocal image stack of two reconstructed neurons (Z+: top left; Z–: bottom left) and the correlating Sholl analysis masks (Z+: top right; Z–: bottom right). (**B**). Quantification of soma surface (**B_1_**) and number of intersections for each Sholl plot (**B_2_**) for 17 Z+ and 15 Z– neurons recorded from 19 *Slc1a6-EGFP* mice. * denotes *p* < 0.05. In B_1_ each triangle represents a data point of a single neuron, in B_2_ each triangle represents the mean of all neurons in a ZebrinII group. See Appendix A for all statistical data. Data are represented as mean ± SEM.

## Data Availability

The data presented in this study are available upon request to the corresponding authors.

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
