# Peer review of "Activity of Cerebellar Nuclei Neurons Correlates with ZebrinII Identity of Their Purkinje Cell Afferents"

_cells, 2021, doi:10.3390/cells10102686_

Round 1

Reviewer 1 Report

    In this manuscript, Beekhof and authors described the differences of firing rates in CN neurons innervating by ZebrinII + or ZebrinII- PCs. The experimental design is straightforward, and the results are clear. Overall, Zebrin-related cerebellar modulation is important and this article is worth publishing. However, there is a concern related to the authors’ interpretation about bursts in CN neurons.

    The authors described increased bursts in Z-CN. However, there was no difference in CV and CV2 between Z- and Z+ CN neurons. Therefore, Z+ and Z- groups have similar firing modulations, with the center of distribution is higher in Z- group. On the other hand, the author defined bursts by an criteria of absolute firing rate, which is 3 consecutive neurons firing >100 Hz. Therefore, it is likely that the Z- group could have more bursts, simply due to the selection criteria of bursts and the effect of ISI distribution. The same observation is also supported by Fig 2, which shows the similar trend of mean firing rates and burst firing rates. Bursting neurons should have post-burst silent period related to activated K+ channels. Therefore, silent period should be considered as a critical criterion for burst detection. It is necessary to re-examine the data before making the conclusion about burst rates.

    The author defined CN neurons into Z+ and Z- groups based on its PC innervating profiles. It would be nice to have a table/figure to summarize the PC innervating profiles of Z+ and Z- CN neurons. It may also help to explain the dissociations, as mentioned by the authors, that differences in PC firings start at P12, but differences in CN firings are not seen until P40.

Reviewer 2 Report

While it is known that cerebellar Purkinje cells (PCs) can be distinguished as Z+ or Z- (based on ZebrinII expression levels), little or none was known on the deep cerebellar nuclei (CN) neurons receiving innervation from Z+ or Z- PCs. In this work, Beekhof and colleagues try to fill the gap by performing extracellular recordings from CN neurons in awake Slc1a6-EGFP mice. Interestingly, their results show that CN neurons spontaneous firing indeed shows an (age-dependent) correlation to the Z+/Z- nature of the PC afferents. Whole-cell patch-clamp recordings in acute slices seem to suggest that this difference does not depend on intrinsic properties of DCN neurons but might rely on synaptic afferents.

Major issues:

1) A key point of the study is whether the difference in CN neurons spontaneous firing is an intrinsic property of the cells or depends on synaptic inputs. The authors performed whole-cell patch-clamp recordings on acute slices to address this issue. Though the recordings are sound, the use of this technique, which causes a dilution of the intracellular medium, might not be ideal. It is known that intrinsic properties as autorhythmic activity might rely on very fragile intracellular mechanisms, which might be impaired by whole-cell-derived perturbation and dilution of the intracellular medium. Then, it is crucial to repeat these experiments using the cell-attached configuration, or perforated-patch, or at least extracellular recordings in slices. The authors can limit the new experimental session to the range of age that is more significant in vivo.

2) In addition to the change in the technique used to address CN neurons intrinsic properties, in the same experiments, I suggest blocking glycinergic transmission, too. This is because local interneurons in the CN are not well characterized, and glycinergic transmission is represented (see, for example, Uusisaari and Knopfel, Cerebellum 2011). This addition will rule out any possible synaptic perturbation in the results obtained.

3) It is important to specify how many neurons were recorded from Slc1a6 mice and C57BL6 mice. Is it possible that the different mouse models could affect the difference observed in the experiments? This should be commented.

Minor issues:

a) It would be helpful to specify in the Methods why the Slc1a6 mouse was chosen.

b) Why was a K-gluconate intracellular solution used for extracellular recordings?

c) Methods line 112: try to quantify the “appeared normal” for the EcoG after the anesthesia.

d) line 125: try to be consistent in the use of numbers as written in letter or not (twenty-two mice here, 62 at line 65)

e) line 125: are these 22 mice a subset of the in vivo cohort?

f) provide a high-resolution version of the figures.

g) Discussion, line 464: anticipate the EAAT4 expression levels used to identify Z+ PC in the Methods section.

h) line 477: The cited figure supplement 2A2 does not exist.

i) I suggest putting all the statistical tables in the supplementary to obtain a more easily readable main text.

l) Conclusions, line 497: the word “behaving” might be misleading. Mice were awake but not entrained in specific behaviors during the recordings.

Reviewer 3 Report

In this paper, Beekhof and collaborators evaluate the activitiy of cerebellar nuclei neurons based on the expression (or lack of expression) of ZebrinII.

Their results indicate that the components of the olivocerebellar loop work at a higher rate in the ZebrinII-negative regions.

The article is well written, the methodology is explained in detail and the bibliography is adequate, although the use of updated references is recommendable.

This work is recommended for publication if some minor concerns, below detailed, are addressed.

1) What is the biological significance of Zebrin II. ZebrinII-expression (or lacking) neurons are more resistant to degenerative processes? The authors must enlarge the description about Zebrin II expression in the cerebellum throughout the introduction.

2) Please increase the resolution of the Figures.

3) Should Panel C in Figure 1 be referred as C1? In table 1 is indicated as Figure C1.

4) It would be very recommendable to show immunohistochemical labelling of ZebrinII in rat cerebellum describing Zebrin II-positive and negative regions.

Round 2

Reviewer 1 Report

The authors have addressed all my questions. I would suggest to accept the manuscript in its current form. 

Reviewer 2 Report

The authors addressed every issue I raised in my previous report.

I have an additional comment about:

3) It is important to specify how many neurons were recorded from Slc1a6 mice and C57BL6 mice. Is it possible that the different mouse models could affect the difference observed in the experiments? This should be commented.

 Answer: In the current study we used 18 Slc1a6 mice and 44 C57BL/6 mice for the in vivo recordings, which were randomly mixed over all age groups. In total 111 out of the 404 reported recordings were from Slc1a6 mice. All 22 mice for the in vitro recordings where Slc1a6 mice. In our view, it is unlikely that the expression of a EGFP tag linked to EAAT4 expression in Purkinje cells (Dehnes et al., 1998, J Neurosci) will lead to a difference in CN spiking patterns when all synaptic transmission is blocked. Nevertheless, we calculated the difference between Slc1a6 and C57BL/6 recordings, but found no significant difference between any parameter. We now report these numbers in the material and methods section on page 3, line 73-82.

---

The comment on the models used referred to the mouse strain of origin for Slc1a6 mice, which seems to be the FVB/NTac, different from the C57BL/6. Mice belonging to different strains can show very different behavior and different cellular properties (see for example: Bothe et al., 2004: Genetic and behavioral differences among five inbred mouse strains commonly used in the production of transgenic and knockout mice). It is common practice to obtain results from mice coming from the same strain of origin. Nevertheless, the authors did not find any difference in the results obtained from the two strains and this is now clearly stated in the text.